# Ethyl Formate Fumigation for Control of the Scale Insect *Asiacornococcus kaki*, a Quarantine Pest on Sweet Persimmon, *Diospyros kaki*: Efficacy, Phytotoxicity and Safety

**DOI:** 10.3390/insects14040341

**Published:** 2023-03-30

**Authors:** Tae-Hyung Kwon, Jin-Hun Cho, Dong-Bin Kim, Gi-Myon Kwon, Ki-Jeong Hong, Yonglin Ren, Byung-Ho Lee, Min-Goo Park

**Affiliations:** 1Institute of Quality and Safety Evaluation of Agricultural Products, Kyungpook National University, 80 Daehak-ro, Daegu 41566, Republic of Korea; 2Komohana Research Center, University of Hawaii, Manoa 875, Hilo, HI 96720, USA; 3Institute of Agriculture and Life Science, Gyeongsang National University, Jinju 52828, Republic of Korea; 4Bio Utilization Institute, Sesamkeingil 83-10, Andong 36614, Republic of Korea; 5Department of Plant Medicine, Sunchon University, 255 Jungang-ro, Suncheon 57922, Republic of Korea; 6College of Science, Health, Engineering and Education, Murdoch University, 90 South Street, Murdoch, WA 6150, Australia; 7Department of Bioenvironmental Chemistry, Jeonbuk National University, Jeonju 54896, Republic of Korea

**Keywords:** quarantine treatment, liquid ethyl formate, methyl bromide alternative, scale, sweet persimmon fruit

## Abstract

**Simple Summary:**

The aim of this study was to evaluate the effectiveness the fumigant of ethyl formate (EF), an alternative to methyl bromide, in controlling the pest *Asiacornococcus kaki* on sweet persimmon fruit. The study assessed the hatching rate of eggs and the survival rates of nymphs and adults of *A. kaki* at low temperatures as well as the lethal concentration of EF required to kill 50% and 99% of its population. Large-scale tests were conducted in a 20 ft reefer container at 5 ± 1 °C for 6 h to confirm the efficacy of EF on different life stages of *A. kaki*. Our results indicated that EF is an effective pesticide for *A. kaki* without causing phytotoxic damage to persimmon fruit. However, at the egg stage, which was the most tolerant, the *A. kaki* population was not completely controlled in linear low-density polyethylene (LLDPE)-packaged fruit. Overall, EF is a promising fumigant for the quarantine pre-treatment of sweet persimmon fruit before it is packed with LLDPE film, specifically to control *A. kaki* located beneath the calyx of the fruit.

**Abstract:**

Sweet persimmons are a valuable export commodity. However, the presence of live insects such as *Asiacornococcus kaki* limits their access to many export markets. Methyl bromide, traditionally used for pest control, is damaging to human health and the environment. Ethyl formate (EF) is a viable alternative; however, its effectiveness against *A. kaki* on sweet persimmon fruit is unknown. We evaluated the effectiveness of EF fumigation in controlling *A. kaki* present under the calyx of persimmon fruit. The hatching rate of eggs and the survival rates of nymphs and adults of *A. kaki* at low temperatures, its LCt_50_ and LCt_99_ after EF exposure, and phytotoxic damage caused by EF were evaluated in laboratory-scale and commercial-scale tests. The dose–response tests showed that the EF LCt_99_ at 5 °C was 9.69, 42.13, and 126.13 g h m^−3^ for adults, nymphs, and eggs, respectively. Commercial-scale tests demonstrated EF efficacy against all *A. kaki* stages without causing phytotoxic effects on persimmons, though the eggs of *A. kaki* were not completely controlled in linear low-density polyethylene (LLDPE)-packaged fruit. This study demonstrated that EF is a potential fumigant for quarantine pretreatment, especially before persimmon fruit is packed with LLDPE film, to control *A. kaki* infesting sweet persimmon fruit.

## 1. Introduction

Sweet persimmon, *Diospyros kaki* Thunberg, is a major temperate fruit crop worldwide, with a total trade value of $608 million as of 2020. Spain is its top exporter, with $269 million exports [1]. In 2020, the Republic of Korea exported 5609 tons of persimmons worth $9.33 million, ranking it the 8th largest exporter worldwide [2]. To comply with the current phytosanitary treatment protocols, when quarantine pests are detected at Korean ports, persimmons exported from Korea are usually fumigated with methyl bromide (MB) [3]. However, MB was designated as a significant ozone-depleting agent in 1992, and its use has been phased out for most applications since 2015 [3,4]. MB also poses a health risk to workers because of its acute or chronic toxicity [5,6,7,8]. Workers experience the functional degradation of the nervous system when exposed to MB concentrations exceeding the permissible limit of 1 ppm [9,10,11]. Therefore, the International Plant Protection Convention recommended the replacement or reduction of MB use as a phytosanitary measure [12].

Current MB alternatives for treating persimmons include spraying insecticides; however, these are insufficient for controlling pests present beneath the calyx of the fruit because they are protected from the insecticide spray [13]. Other treatments, such as hot water treatment, modified atmosphere, and hot and cold storage, are also insufficient for effective control [14,15,16]. Ethyl formate (EF) was developed as a fumigant for dried fruit in 1925 and has been successfully used on fruits and certain stored grains in India and Australia since the 1980s [17]. Liquid EF mixed with nitrogen carrier gas was developed to rid of *Gonipterus platensis* (Eucalyptus weevil) from export apples in Australia [18] and treat mealybugs in imported citrus in South Korea [19]. EF can also treat a wide range of grains and cereals [20,21] and has been applied for the post-harvest treatment of fruits and vegetables [19,22,23]. The advantages of using EF as an MB alternative are short fumigation holding periods, low toxicity in mammals, rapid breakdown to ethanol and formic acid, and minimal to no adverse effects on the environment [21,24]. The Food and Drug Administration also reviewed its use as a flavoring agent and characterized it as “generally recognized as safe” [25]. However, EF does not easily penetrate inside the package owing to its relative low vapor pressure [26]. 

Various pests, including mites, moths, and scales, are associated with persimmons for export [27]. These pests are typically found beneath the large protective calyx of the fruit; hence, controlling them is problematic yet essential for establishing and maintaining persimmon export markets [28,29]. Most of these species are not classified as quarantine pests. However, *Asiacornococcus kaki* Kuwana in Kuwana & Muramatsu (Hemiptera: Coccoidea), a major pest of the fruit, is considered a quarantine pest in the export of sweet persimmons to certain countries [30]. 

In this study, the feasibility of using EF as an alternative to MB for pest treatment in persimmons was evaluated as follows: (1) the pests present beneath the calyx of sweet persimmons were identified; (2) the efficacy of the EF treatment on *A. kaki*, the pest with the highest occurrence rate in the identification, was evaluated in small-scale laboratory trials; (3) the effects of the EF treatment on sugar content, hardness, and external damage to persimmons were assessed in small-scale laboratory trials; (4) the efficacy of EF on all stages of *A. kaki* was assessed in packaged and unpackaged conditions combined with an evaluation of phytotoxicity in commercial-scale trials using 28 m^3^ shipping containers; and (5) the concentrations of EF desorbed from treated persimmons were monitored under simulated storage conditions in a shipping container for 24 h after completion of fumigation. In this study, liquid EF was used with nitrogen gas for safe and easy handling [19,31].

## 2. Materials and Methods

### 2.1. Insects and Fumigant

Eggs, nymphs, and adults (about one week age) of *A. kaki* (persimmon scale) used for the bioassays were collected from a persimmon orchard in Andong, South Korea between August and September, 2020–2021 (GPS 36.692965, 128.737978). The insects were indentificated by an expert of scale in Korea. Fumate^®^ (>99% pure liquid EF) was supplied by Safefume Inc., Hoengseong, South Korea. Commercial-scale fumigation trials were conducted in 20 ft refrigerated shipping containers (reefer containers with an internal volume of 28 m^3^). Liquid EF was vaporized using a commercial EF vaporizer (SFM1; Safefume Inc., Hoengseong, South Korea) and mixed with nitrogen carrier gas to form a nonflammable EF fumigant formulation. 

### 2.2. Investigation on Pest Species on Sweet Persimmon Fruit

Sweet persimmon fruits were harvested from four conventional persimmon orchards in San-Cheong, Chang-won, Mil-yang, and Jin-ju, South Korea. Forty fruits were randomly collected from ten trees in each orchard. Arthropod species present beneath the calyx were investigated using a microscope (Stemi DV4; Zeiss Korea, Seoul, Republic of Korea). The number of species present was counted and recorded by an expert entomologist in Animal and Plant Quarantine Agency (APQA).

### 2.3. Effectiveness of EF against A. kaki in Small-Scale Laboratory Trials

Laboratory fumigation trials with EF in *A. kaki* and its phytotoxicity in sweet persimmons were evaluated. A glass desiccator (Duran^®^ 6.9 L; DWK Life Science, Mainz, Germany) equipped with a gas-sampling port was used as the fumigation chamber. A fan was placed at the bottom of the chamber to stir the air inside and ensure the even distribution of the fumigant. 

Prior to fumigation, approximately 30 eggs, nymphs, and adult insects were inoculated in the calyx of persimmon and placed inside separate insect-breeding dishes (Ø = 90 mm) in fumigation chambers loaded with approximately 20% (loading ratio) of 4–5 sweet persimmon fruits. The chambers were sealed and moved to a temperature-controlled room (5 ± 2 °C) overnight. Vaporized EF was injected into the fumigation chambers using a gas-tight syringe (100 or 500 mL, SGE). The dosage was calculated according to the equation reported from Ren et al. (2011) [32]. Dosages of EF were 5.0, 13.0, 21.2, 29.8, 37.1, 43.8, 45.6, 73.5, 80.5, 86.3, and 90.1 g m^−3^ for eggs; 4.8, 11.0, 18.5, and 24.8 g m^−3^ for nymphs; and 1.6, 5.6, 6.4, and 7.6 g m^−3^ for adults. Each dosage was replicated three times. Desiccators were placed in a temperature-controlled room at 5 ± 2 °C for 6 h. After fumigation for 6 h, the desiccators were opened and aired for 1 h in a fume hood. The insects were removed from the desiccators and reared at 25 ± 2 °C with 55 ± 5% relative humidity (RH). The endpoint of mortality was determined after 24 h for adults and nymphs and after 8 days for eggs. 

### 2.4. Phytotoxicity of EF on Sweet Persimmon Fruit in Small-Scale Laboratory Trials

The phytotoxicity of treated (>10 number) sweet persimmon fruits was evaluated against untreated fruits after fumigation and storage for 7 d and 14 d at 5 ± 1 °C and 55 ± 5% RH. The parameters measured were fruit firmness, sugar content, the color of the fruit surface, and external damage. Firmness was measured using a fruit firmness tester (53205 digital fruit firmness tester; TR Turoni, Forli, Italy) equipped with an 8 mm steel plunger. The fruits were compressed to 6 mm at the equatorial zone at a rate of 0.5 mm s^−1^, and the maximum number generated during the test was recorded. The firmness test was repeated three times per fruit, and ten fruits were analyzed per treatment and control. The soluble sugar content was analyzed using a portable refractometer (Hand refractometer ATC-1E; Atago Co., Ltd., Tokyo, Japan). The fruit samples were individually ground in a tissue grinder and filtered through a funnel covered with filter paper (Whatman No. 4; All for lab, Seoul, Republic of Korea). The resulting clear solution was collected for the analysis. The liquid (0.5 mL) was dropped onto the refractometer, and the sugar content was recorded. Ten fruits were tested per treatment. The color change on the fruit surface was measured using a colorimeter (SpectroDens; Techkon GmbH, Konigstein, Germany). Three circles (ϕ = 10 mm) were randomly marked on the fruit at their equatorial zone, and the color within the circles was measured. These readings were expressed as Hunter *L**, *a**, *b** values. In this study, skin redness was represented as an a/b ratio [33]. The development of spots and other visible symptoms of external damage to the fruit were scored subjectively as 0 (none affected), 1 (slight, <5% affected), 2 (moderate, <25% affected), or 3 (severe, >25% affected). 

### 2.5. Commercial-Scale Trials in a 20 ft Reefer Container

Two scale-up trials on export persimmon fruits were performed in a 20 ft (28 m^3^) reefer container located in the Plant Quarantine Technology Center. EF concentrations of 35 or 50 g m^−3^ were introduced into the container via a vaporizer with nitrogen carrier gas and fumigated at 5 ± 1 °C for 6 h. The dose of 35 g m^−3^ was selected because this is the dose used for disinfecting banana imported in Korea, and the dose of 50 g m^−3^ was selected to guarantee 99% mortality of *A. kaki*. The sweet persimmon fruit were loaded with a 20% loading ratio inside the container. Tested insects were placed in an insect breeding dish using the same method as that of the small-scale laboratory trials and placed inside boxes using two methods, unpacked and linear low-density polyethylene (LLDPE)-packaged fruit, to check the permeability of EF gas through the LLDPE film. The film was 0.06 to 0.08 mm thick. The concentration of EF was determined using an Agilent Technology 7890N gas chromatograph (GC) equipped with a flame ionization detector (FID) after performing isothermal separation on a 30 m × 0.32 mm I.D (0.25 µm film)-fused silica capillary HP-5 column (J&W 19091 J-413; Aglient, Santa Clara, CA, USA). Gas-monitoring tubes were placed in five positions (front down and up, middle up, rear down, and up) inside and outside the packages to measure the gas concentration. EF in the reefer container was sampled by withdrawing gas using an air pump, and its concentration was analyzed using GC-FID. After fumigation for 6 h, the container was opened and ventilated for 24 h before the desorption rate was measured by GC-FID. The fruits were transported to a laboratory for evaluating the disinfestation efficacy and phytotoxicity of EF. The firmness and external damage were measured. After 3 d of incubation, the mortality of adult and nymph-stage insects was assessed under a microscope. After 7 d of incubation, the hatching rate of eggs was investigated. After 14 d of storage, phytotoxic damage to sweet persimmons was evaluated. The firmness of the fruit and external damage were measured as described in Section 2.4.

### 2.6. Fumigant Concentration and Ct (Concentration × Time) Product

During fumigation, the concentration of EF was monitored using GC-FID. The GC oven, injector, and detector temperatures were 150, 240, and 240 °C, respectively. Helium was used as the carrier gas at a flow rate of 2 mL min^−1^. 

The concentration of EF was calculated based on the GC-FID peak area against external EF gas standards and was calibrated periodically using external gas standards. For laboratory and commercial-scale trials (APQA site, Gimcheon, Korea), gas standards were prepared by extracting a calculated amount of air from a 1 L Erlenmeyer flask (Cat. No. FE 1 L/3; Bibby Sterilin, Staffordshire, UK) containing 5–6 glass beads (Ø = 3–5 mm) equipped with a cone/screw-thread adapter (Quickfit, STS; Bibby Sterilin) and injecting a calculated amount of liquid EF into the Erlenmeyer flask. The accurate volume of each Erlenmeyer flask and inlet system was measured and calculated from the weight of water required to fill the container at 20 °C. 

Liquid EF was transferred using a 10 µL syringe (Cat. No. 10R-GT; SGE, Melbourne, Australia), and a 100 µL airtight syringe with a valve (Cat. No. 005279; SGE) was used for injecting gas samples into the GC. 

To calculate the Ct product, the concentration of EF in the fumigation chamber was determined at time intervals of 0.5, 2.0, and 6.0 h after injection of EF with three replications. The Ct values were calculated as previously described [32].

### 2.7. Statistical Analysis

The toxicological dose response to EF by *A. kaki* was analyzed using Probit analysis [34], based on a computer program written by Ge Le Pattourel, Imperial College, London and adopted by Don-Pedro [35]. The slopes of the probit transformations were determined along with Chi-square tests of data homogeneity for different treatments and developmental stages of *A. kaki*. The indices of toxicity derived from these analyses were LCt_50_ = median lethal concentration that causes a 50% response (mortality) and LCt_99_ = lethal concentration that causes a 99% response (mortality) to exposed *A. kaki*. At least ten different Ct products were tested to ensure that the observed data covered mortality rates from 0 to 100% and included the intermediate range adequately. All data-related assessments of phytotoxicity, such as firmness, sugar content, surface color change, and external phytotoxic damage index, were analyzed using the software package Minitab version 10.1. (Minitab Inc., State Colleage, PA, USA). 

## 3. Results

### 3.1. Identification of Invertebrate Species 

Forty sweet persimmon fruits were tested for invertebrate infestation. A total of 588 individual invertebrates were present beneath the calyxes the 40 examined fruits. These included *A. kaki* (348), *Tetranychus urticae* (two-spotted spider mites; 117), *Amblyseius eharai* (119), and others (4). The most dominant species were *A. kaki* (59.18%), *A. eharai* (20.24%), and *T. urticae* (19.90%). Other species were found in low numbers (<1.0%) (Figure 1).

### 3.2. Efficacy of EF in Controlling A. kaki in Small-Scale Laboratory Trials

The L(Ct)_50_ and L(Ct)_99_ values of EF for *A. kaki* were 41.70 and 126.13 g h m^−3^ for the egg stage, 6.67 and 42.13 g h m^−3^ for the nymph stage, and 3.63 and 9.69 g h m^−3^ for the adult stage, respectively. These were determined from the fitted slopes of 4.84, 2.91, and 5.45, for the egg, nymph, and adult stages, respectively (Table 1). The egg and adult stages were most tolerant and sensitive to EF treatment, respectively.

### 3.3. Phytotoxicity of EF on the Fruit in Small-Scale Laboratory Trials

The firmness values of sweet persimmon after 7 and 14 d were 3.44 and 2.29 kg force cm^−2^ (kgf) for untreated fruits and 3.57 and 2.22 kgf for treated fruits, respectively. There were no significant differences in firmness between untreated and treated samples (*p*-value > 0.05). The mean soluble sugar content after 7 and 14 d was 13.6 and 14.4% for untreated (control) fruits, 13.7 and 14.6% for 35 g m^−3^ EF treatment, and 14.4 and 14.6% for 50 g m^−3^ EF treatment, respectively. There was no significant difference in the soluble sugar content between the untreated and treated samples (*p*-value > 0.05). There were no changes in color or visible external damage as a result of these two EF treatment dosages (30 and 50 g m^−3^). After 7 and 14 d, the redness ratios of fruits were 0.32 and 0.28 for untreated fruit, 0.29 and 0.26 for 35 g m^−3^ EF treatment, and 0.30 and 0.29 for 50 g m^−3^ EF treatment, respectively (Table 2). These results indicated that EF fumigation of less than 50 g m^−3^ for 6 h caused no phytotoxic damage to sweet persimmon fruits.

### 3.4. Commercial-Scale Trials

During the 6 h fumigation holding period in the 20 ft reefer container, the concentration of EF inside the fruit boxes decreased because of adsorption on the fruit. The EF concentration at the end of the fumigation period was 30% of the initial dose (Figure 2). The EF concentration inside LLDPE-packaged fruits slowly increased during the first 4 h of the holding period because EF penetrated through the LLDPE film and into the packaged fruits, achieving equalization. The calculated *Ct* product in the trial fumigated with 35 g m^−3^ of EF was 80.2, 44.4 g h m^−3^ for unpackaged and packaged fruits, respectively, which was higher than the L(Ct)_99_ values of EF on nymph and adult stages of *A. kaki* but lower than that for the egg stage. *A. kaki* eggs were not completely controlled, while the nymphs and adults were completely killed. When treated with 50 g m^−3^ EF, all stages of *A. kaki* were completely controlled in the unpackaged fruits, although the efficacy of EF in controlling the egg stage of *A. kaki* in LLDPE-packaged fruit was 74.9%. This was attributed to the *Ct* product of 130.1 and 60.6 g h m^−3^ in the unpackaged fruits and packed fruits, respectively (Table 3). There were no significant differences in the quality of the persimmon fruits, firmness, and external damage between the control fruits and fumigated fruits at 14 d after fumigation.

During the aeration period, the EF concentration in LLDPE-packaged fruit decreased more slowly than that in unpackaged fruits (Figure 3). The EF concentration in unpackaged fruits decreased to <100 ppm (TLV-TWA of EF) within 1 h; however, more than 15 h was required for the same in LLDPE-packaged fruits, which could pose an acute inhalation risk to workers. Therefore, to ensure worker safety, ventilation should be maintained for >1 h and >15 h for unpacked and LLDPE-packaged fruits, respectively.

## 4. Discussion

EF treatment was effective against all stages of the quarantine invertebrate pest *A. kaki*, including the egg stage that was the most tolerant and present beneath the calyx of sweet persimmon fruits. Eggs and other life stages of *A. kaki* were also successfully controlled on unpacked sweet persimmon in a commercial-scale 20 ft reefer container. For the treatment of export sweet persimmon at 5 °C, the optimum level of fumigation was 50 g m^−3^ EF for a 6 h holding period. This resulted in no phytotoxic damage to the fruits and did not have any impact on worker safety after 15 h of aeration since EF decreased to <100 ppm (TLV-TWA of EF).

In a previous study, the adult stage of *Pseudococcus maritimus* (grape mealybug) was the most susceptible to EF treatment, while its egg stage was the least susceptible [36]. *P. maritimus* crawlers and adults have been successfully controlled with 3–4% EF treatment for 1 h (100–125 g h m^−3^), whereas 4.9% EF was required to control its eggs [37]. Adults of *Planococcus citri* (citrus mealybug) were found to be more susceptible to ethyl formate and methyl bromide fumigation compared with its eggs [31]. *Frankliniella occidentalis* (western flower thrips) was completely controlled on strawberries with 79.4 g m^−3^ of EF fumigation for 1 h, while the same concentration controlled *T. urticae* only partially (66%) [38]. A 12 h fumigation with 10 g m^−3^ of EF was sufficient to control diapausing *T. urticae* in their adult stage [39], whereas a 6 h fumigation with 10 g m^−3^ completely controlled them in field trials [40]. These reports indicate that EF can be applied to control mites and certain insects effectively. However, not much is known about the efficacy of EF on *A. kaki*, especially at low temperatures. In this study, the LCt_99_ of EF on eggs, nymphs, and adults of *A. kaki* were 126.13, 42.13, and 6.67 g h m^−3^, respectively, indicating that eggs were the most tolerant stage towards EF treatment (Table 1). This is consistent with the results found for other mealybugs [31,36,37]. This treatment could potentially completely control other invertebrate pests, such as mites or thrips [38,39,40].

Nine invertebrate species were described as pests of sweet persimmon in sorting areas, including those for export, in 2011 and 2012, in Korea [27]. Among them, *Panonychus ulmi* (European red mite), *Ponticulothrips diospyrosi* (Japanese gall-forming thrips), and *A. kaki* have been listed as quarantine pest [30]. Mites of *A. eharai* and *T. urticae* were the most frequently found, which have not been classified as quarantine pests. The remaining pests, including *A. kaki*, were rarely found. However, *A. kaki* was the most frequently found pest on persimmon fruit in our study (Figure 1). This difference in pest occurrence could be related to the different areas investigated between the two studies.

The phytotoxicity of EF was assessed for various fruits such as bananas, oranges, grapes, and persimmons, and it was suggested that EF does not affect the quality of those fruits [19,31,40,41]. In this study, there was no significant change in firmness, sugar content, and redness ratio of persimmons and no evidence of external damage to persimmons 14 d after the EF treatment with 35 and 50 g m^−3^ at 5 ± 1 °C in the laboratory trials (Table 2). Phototoxicity was not observed in the commercial-scale trials with 28 m^3^ shipping containers at the same storage period and temperature. This result is consistent with the findings of previous studies [19,31,40,41].

The penetration of fumigants depends on packaging of the commodity, which is critical for achieving LCt_99_ to disinfect the target pests [32,42]. EF does not effectively penetrate plastic bagging used in banana cartons during commercial shipping, which reduces the mortality of *P. citri* eggs [31]. This study demonstrated the similar efficacy of EF on pests inside packaging, where EF treatment for LLDPE-packaged sweet persimmon fruit was insufficient to kill eggs (Table 3, Figure 2). The incomplete control of the egg stage of *A. kaki* was expected under the packaged conditions as the *Ct* product was 44.4 and 60.0 g h m^−3^ with dosage of 35 and 50 g m^−3^, respectively; this was markedly lower than the LCt_99_ of the pest (126.13 g h m^−3^). By contrast, unpacked fruit with a dosage of 50 g m^−3^ achieved LCt_99_, resulting in complete mortality of all stages of *A. kaki*, including the eggs.

The fumigant absorbed during fumigation is desorbed soon after opening a fumigation enclosure, increasing the occupational and environmental risks for fumigation workers. In a previous study on pest control in oranges, the concentrations of EF and MB fluctuated between 12.6–36.6 ppm and 9.4–32.5 ppm, respectively, during the 24–72 h post-fumigation period [19]. This indicated that the EF levels were below the permissible exposure limit (PEL) of 100 ppm, while MB levels were above the PEL of 1 ppm. Consistent with this, in the present study we observed that EF levels decreased to below the PEL of 100 ppm approximately 15 h after the initiation of degassing (Figure 3).

A limitation of this study was the insufficient pest numbers in for large-scale trials for determining the efficacy for quarantine areas. Future research should assess EF efficacy on a larger number of pests and include a variety of internal pests, such as moths. Nevertheless, this study identified the *Ct* product required to control *A. kaki*, the most dominant species and quarantine pest in sweet persimmons. The *Ct* product was confirmed to be applicable in commercial-scale trials without causing phytotoxicity to the persimmons. Additionally, this study demonstrated that EF is safe for workers during ventilation periods after fumigation. Thus, we suggest that EF is a potential alternative to MB treatment for controlling *A. kaki* on persimmons during quarantine pre-treatment associated with trade.

## Figures and Tables

**Figure 1 insects-14-00341-f001:**
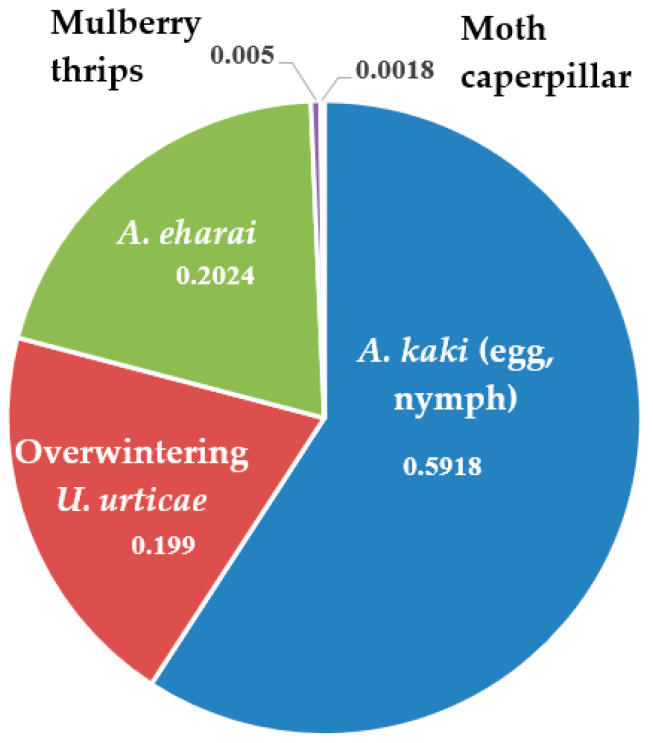
Invertebrate species present beneath the calyx of sweet persimmon fruit after harvest (total insects = 588).

**Figure 2 insects-14-00341-f002:**
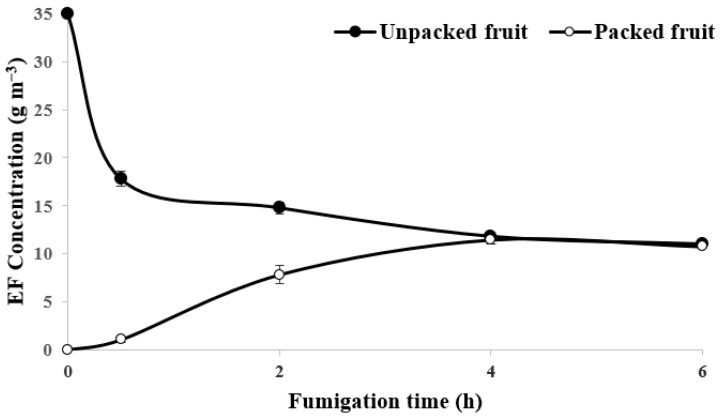
Sorption pattern of ethyl formate on unpacked and LLDPE-packed sweet persimmon fruit in a 20 ft (28 m^3^) reefer container during 6 h of fumigation.

**Figure 3 insects-14-00341-f003:**
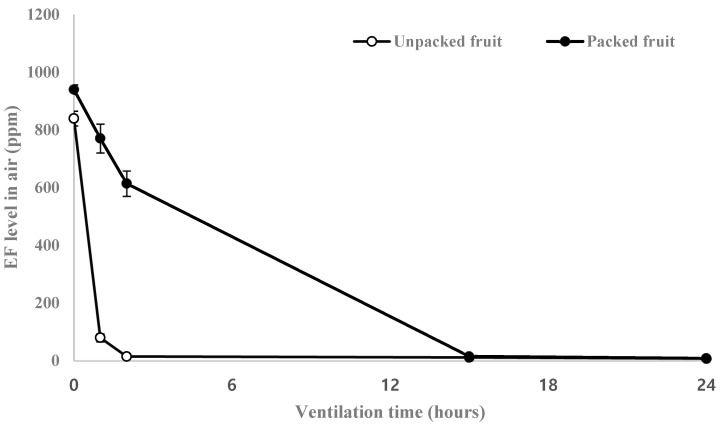
Desorption of ethyl formate from unpacked and LLDPE-packed sweet persimmon fruit in a 20 ft (28 m^3^) reefer container at different ventilation times.

**Table 1 insects-14-00341-t001:** Effectiveness of ethyl formate against all stages of *Asiacornococcus kaki* at 5 ± 1 °C on sweet persimmon fruit.

A. *kaki* Stage	Number of Individuals Treated	Concentration × Time Products (g h m^−3^)		Slope ± SE ^a^	Df ^b^	χ^2^
		L(Ct)_50_ (95% CL)	L(Ct)_99_ (95% CL)			
Egg	1020	41.70 (38.17–45.22)	126.13 (105.91–161.44)	4.84 ± 0.44	31	120.23
Nymph	340	6.67 (5.87–7.57)	42.13 (31.27–64.07)	2.91 ± 0.22	10	38.2
Adult	340	3.63 (2.82–4.29)	9.69 (8.52–11.48)	5.45 ± 0.56	10	62.3

^a^ Standard Error. ^b^ Degrees of freedom.

**Table 2 insects-14-00341-t002:** Phytotoxicity assessment of sweet persimmon fruit exposed to 35 and 50 g m^−3^ of ethyl formate for a 6 h exposure period at 5 ± 1 °C.

Dosage of Ethyl Formate (g m^−3^)	Ct Products (g h m^−3^)	Fumigation Time (h)	Storage Period (Days)	Firmness ± SE ^1^ (kgf)	Sugar Content ± SE (%)	Redness Ratio ^2^	External Damage ^3^
Untreated	-	-	7	3.44 ± 0.32 a	13.6 ± 0.3 a	0.32 a ^4^	0
			14	2.29 ± 0.44 a	14.4 ± 0.1 a	0.28 a	0
35	85.99	6.0	7	3.57 ± 0.31 a	13.7 ± 0.2 a	0.29 a	0
			14	2.22 ± 0.65 a	14.6 ± 0.2 a	0.26 a	0
50	131.13	6.0	7	3.77 ± 0.22 a	14.4 ± 0.2 a	0.30 a	0
			14	3.43 ± 0.09 a	14.6 ± 0.3 a	0.29 a	0

^1^ Standard error. ^2^ Redness ratio = Hunter *a**/*b**. ^3^ Damage score: 0 (none), 1 (slight), 2 (moderate), and 3 (severe). ^4^ Same letter after values means not significantly different at *p* > 0.05.

**Table 3 insects-14-00341-t003:** Commercial-scale fumigation trials with ethyl formate for control of *Asiacornococcus kaki* and assessment of phytotoxicity on sweet persimmon fruit at 5 ± 1 °C for 6 h exposure period in a 20 ft reefer container.

Ethyl Formate Dose	Package Types	Ct Products (g h m^−3^)	No. *A. kaki* Eggs		Mortality (%)	No. *A. kaki* Nymphs		Mortality (%)	No. *A. kaki* Adults		Mortality (%)	Firmness (Kg Force cm^−2^) (kgf)	Damage Score ^1^
			Total	Unhatched		Total	Alive		Total	Alive			
Untreated	LLDPE	-	153	6	3.9	126	126	-	27	27	-	3.80 a ^2^	0
	Unpacked	-	98	9	9.2	183	183	-	40	40	-	3.76 a	0
35 g m^−3^	LLDPE	44.4	246	135	54.9	481	0	100	163	0	100	3.89 a	0
	Unpacked	80.2	187	164	87.7	456	0	100	157	0	100	3.62 a	0
50 g m^−3^	LLDPE	60.6	311	233	74.9	356	0	100	223	0	100	4.11 a	0
	Unpacked	130.1	269	269	100	411	0	100	244	0	100	4.09 a	0

^1^ Damage scores: 0 (none), 1 (slight), 2 (moderate), and 3 (severe). ^2^ Same letter after values means not significantly different at *p* > 0.05.

## Data Availability

All of the data supporting the findings of this study are available from the corresponding author upon reasonable request.

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
