# Peer review of "Ethyl Formate Fumigation for Control of the Scale Insect Asiacornococcus kaki, a Quarantine Pest on Sweet Persimmon, Diospyros kaki: Efficacy, Phytotoxicity and Safety"

_insects, 2023, doi:10.3390/insects14040341_

Round 1

Reviewer 1 Report

Dear authors

Their study is interesting, however needs to be improved...The methodology and result should be written of sequential form. see suggestions inside manuscript

Reviewer 2 Report

In this manuscript the evaluation of ethyl formate fumigation for managing the different life stages of the quarantine pest Asiacornococcus kaki in Diospyros kaki (Sweet Persimmon) was reported. The multiple objectives of the study aimed at the evaluation of the feasibility of using EF as an alternative to MB for pest treatment in persimmons. The study was well planned and it consists of a comprehensive important document on the positive features of EF as a quarantine fumigant.

General comments:

The objective of this work has been clearly defined. The authors of this study reported very interesting information on the efficacy of EF in controlling all life stages of A. kaki. This study serves as a preliminary data source to completely control the pest's life stages.

Although the author mentions that a larger number of pests should be tested to assess EF efficacy, under commercial conditions, to confirm results, and to obtain complete control on the egg stage further future research would be necessary.

Authors used the term g h m-3 as CT product. The term is correct for scientific purposes and it is useful for the comparison of efficacy. However, it means very little to the reader to understand the actual dosages used in each specific fumigation. Therefore it is suggested that where relevant to the judgment of the authors, to mention " Ct product was XX g h m-3 329 with the dosage of YY g m3…". Adding the actual dosage used in each experiment will increase the comprehensibility of the study.

Specific comments:

Line 126:  the term "… after treatment…" is not clear. May be changed to "exposure time"?

Lines 202-204: the sentence "… There were determined from a range of at least ten different Ct products to ensure that the observed data covered mortality from 0 to 100% and adequately covered the intermediate range…" should be rephrased

Line 229: "…Kg…" should be "…kg…"

Line 254: "…m-3 of EF was 80.2, 44.4 g h m-3 for unpackaged fruits, respectively,…" there should be an additional package, maybe "…m-3 of EF was 80.2, 44.4 g h m-3 for unpackaged and packaged fruits, respectively,…"?

Reviewer 3 Report

This work is interesting and clearly written. Meets all requirements for scientific publication. This MS presents interesting results on the use of ethyl formate as fumigant for postharvest control of Asiacornococus kaki, on stored Sweet Persimmon in a laboratory and semi field study. Furthermore, the study describes the absorption of ethyl formate on unpacked and LLDPE-packed sweet persimmon in semi field test. The presented results are of practical use in the post-harvest protection of sweet persimmon. The article is clearly and clearly written.

I have following comments that should be addressed before publishing the MS.

Keywords:

I recommend replacing some repetitive words in the keywords and in the MS title. Different words should be used in the keywords than in the title.

Introduction:

The Introduction is clearly and comprehensibly written. I have no comments on this part.

Material and Methods:

2.4 Effectiveness of EF against A. kaki in small-scale laboratory trials

The methodology is not clearly described. The dosage range of 0.5 to 70.0 g m-3 and three repetitions is not clear. How many repetitions was each dose? I recommend adding the dosage and the number of repetitions for each dosage. I also recommend adding information about the number and times of EF concentration measurements.

2.6 Commercial-scale trials in a 20 ft reefer container

I recommend supplementing the thickness of the LLDPE packaging used.

I recommend adding the times and number of EF concentration measurements

Results:

Line 209: A total of 548 individual - the given number of the total number of individuals does not correspond to the number of given individuals in Figure 1 - the number given here is 584 - please correct it.

Table 3 - part of the assessment of the effectiveness of ET on eggs is not very understandable.

The evaluation was done on the basis of egg hatching as indicated in the methodology. Why mortality is not evaluated in untreated (control) samples? Isn't the mortality rate wrong for the treated samples? For example - dose 35 g*m3 and LLDPE - total number of eggs 246 and hatched 135, the mortality cannot be 54.9% as shown in the table, but 45.1%. The mortality was not 135 eggs, but 111 eggs (246-135).

Please explain this incomprehensible part.

Table 1 - in the methodology it is stated that the effectiveness was verified in one test on 30 eggs, nymphs and adults - why are the numbers of tested individuals different for eggs and nymphs with adults?

I recommend explaining.

Discussions:

The scope of the discussion is adequate and includes the findings of the study. I have no comments on this part.
